# Mechanism of *Takifugu bimaculatus* Skin Peptides in Alleviating Hyperglycemia in Rats with Type 2 Diabetic Mellitus Based on Microbiome and Metabolome Analyses

**DOI:** 10.3390/md22080377

**Published:** 2024-08-22

**Authors:** Min Xu, Bei Chen, Kun Qiao, Shuji Liu, Yongchang Su, Shuilin Cai, Zhiyu Liu, Lijun Li, Qingbiao Li

**Affiliations:** 1College of Ocean Food and Bioengineering, Jimei University, Xiamen 361021, China; xumin1315@foxmail.com; 2Key Laboratory of Cultivation and High-Value Utilization of Marine Organisms in Fujian Province, Fisheries Research Institute of Fujian, Xiamen 361013, China; chenbeifjfri@foxmail.com (B.C.); qiaokun@qiaokun@xmu.edu.cn (K.Q.); cute506636@163.com (S.L.); suyongchang@stu.hqu.edu.cn (Y.S.); caishuilin@hqu.edu.cn (S.C.)

**Keywords:** peptides of *Takifugu bimaculatus* skin, type 2 diabetes mellitus, dipeptidyl peptidase-IV, hypoglycemia, gut microbiota, non-targeted metabolome

## Abstract

In this study, we aimed to explore the hypoglycemic effects of a hydrolysate on *Takifugu bimaculatus* skin (TBSH). The effect of the dipeptidyl peptidase-IV (DPP-IV) inhibitory activities from different TBSH fractions was investigated on basic indexes, gut hormones, blood lipid indexes, viscera, and the gut microbiota and its metabolites in rats with type 2 diabetes mellitus (T2DM). The results showed that the <1 kDa peptide fraction from TBSH (TBP) exhibited a more potent DPP-IV inhibitory effect (IC_50_ = 0.45 ± 0.01 mg/mL). T2DM rats were induced with streptozocin, followed by the administration of TBP. The 200 mg/kg TBP mitigated weight loss, lowered fasting blood glucose levels, and increased insulin secretion by 20.47%, 25.23%, and 34.55%, respectively, rectified irregular hormonal fluctuations, lipid metabolism, and tissue injuries, and effectively remedied gut microbiota imbalance. In conclusion, TBP exerts a hypoglycemic effect in rats with T2DM. This study offers the potential to develop nutritional supplements to treat T2DM and further promote the high-value utilization of processing byproducts from *T. bimaculatus*. It will provide information for developing nutritional supplements to treat T2DM and further promote the high-value utilization of processing byproducts from *T. bimaculatus*.

## 1. Introduction

*Takifugu bimaculatus* (*T. bimaculatus*) is a pufferfish species belonging to the Tetraodontidae family that is renowned for its delicate and flavorful meat. Owing to a series of technological breakthroughs in recent years worldwide, cultivated *T. bimaculatus* has been described as an important edible fish [1,2,3]. *T. bimaculatus* skin, constituting ~27% of its total wet weight, is typically discarded owing to its small spines and coarse taste. In recent years, the hydrolysates of puffer fish skin have exhibited a wide range of nutraceutical, pharmaceutical, and cosmeceutical activities, including antihypertensive, antimicrobial, and matrix metalloproteinase inhibitory activities [4,5,6]. Therefore, the other physiological activities of *T. bimaculatus* skin hydrolysate (TBSH) need to be studied.

Type 2 diabetes mellitus (T2DM) is a significant endocrine and metabolic disorder that can lead to severe complications in various organs [7]. According to the data published by the International Diabetes Federation, the global diabetic population is projected to reach 700 million by 2045, with T2DM representing over 90% of cases [8].

Moreover, patients with coronavirus disease 2019 (COVID-19) with T2DM exhibit significantly higher rates of morbidity and mortality than non-diabetic individuals, presenting a notable challenge to public health and healthcare expenses [9]. Currently, antidiabetic therapies for T2DM are linked to various side effects that affect individuals’ quality of life, such as gastrointestinal discomfort, hypoglycemia, liver and kidney impairment, and fluctuations in blood glucose levels [10]. Hence, more emphasis is placed on diversified treatment elements combined with nutritional intervention during the modern clinical treatment of diabetes to achieve better treatment results. Food-derived peptides have specific advantages in controlling diabetes due to their good safety and easy digestion and absorption [11]. In general, fish skin has a high content of proline, which is a potential precursor protein of the dipeptidyl peptidase-IV (DPP-IV) inhibitory peptide [12]. DPP-IV inhibitors are among the key strategies used for hypoglycemia, as they prolong the half-life of glucagon-Like Peptide-1 (GLP-1) and promote insulin secretion [13]. We previously found that *T. bimaculatus* skin contained a higher proline content, with the collagen yield reaching 49.83% ± 1.85% in dry weight, surpassing the extraction yields obtained from pufferfish by Nagai (10.7%) and pig skin by Jia (38.9%), indicating its potential inhibitory activity against DPP-IV [14]. In the present study, we attempted to investigate the DPP-IV inhibitory and hypoglycemic activities of TBSH.

The structure of the gut microbiota is important in relieving diabetes, obesity, and non-alcoholic fatty liver [15]. DPP-IV inhibitory peptides from various protein sources (camel milk, casein, donkey blood, and so on) can suppress glycemia by regulating the structure and abundance of the gut microbiota [16]. Furthermore, a recent report showed that DPP-IV inhibitory peptides regulate insulin secretion through GLP-1, and this effect appears to depend on the gut microbiota [13]. Similarly, Zhang et al. found that the supplementation of camel milk, which harbored potential DPP-IV inhibitory activity, can increase the abundance of *Allobaculum* in T2DM mice [17]. Given that the gut microbiota plays a key role in the development of T2DM, it is necessary to study its role in the regulation of the host glucose metabolism by food-derived peptides.

Therefore, we evaluated the impact of a peptide fraction from TBSH on the activity of DPP-IV in vitro, assessed the hypoglycemic impact of TBP by elucidating the hyperglycemia-associated characteristics of rats with T2DM, and determined the variations in the gut microbiota and their metabolites. The results of this study will provide information for developing nutritional supplements to treat T2DM and further promote the high-value utilization of processing byproducts from *T. bimaculatus*.

## 2. Results

### 2.1. Characterizing TBSH

The molecular weight distribution of TBSH was ascertained using high-performance gel-filtration chromatography. Peptides with a molecular weight of <1 kDa (TBP) constituted approximately 71.35% (Appendix A). Subsequent to ultrafiltration, the DPP-IV inhibition activity of TBP (Mw < 1 kDa) had the lowest IC50 value (0.45 ± 0.01 mg/mL), while TBP1 (Mw ≥ 1 kDa) had an IC_50_ value of 2.82 ± 0.09 mg/mL and TBSH had an IC_50_ value of 4.78 ± 0.16 mg/mL (Appendix A). Therefore, the hypoglycemic effects of TBP on T2DM rats were investigated.

### 2.2. Effects of TBP on Basic Indexes in Rats with T2DM

The treatment protocol is depicted in Figure 1A. Figure 1B illustrates that the body weights of the T2DM groups decreased, except for the NC group. Furthermore, the blood glucose levels in the TBPL (TBP, 200 mg/kg) and TBPH (TBP, 400 mg/kg) groups decreased and were significantly different from those in the M group; the decrease was more significant in the TBPH group (Figure 1C, *p* < 0.05). The blood glucose levels during 0–120 min and the area under the curve (AUC) in the TBP groups were markedly higher than those in the M group, especially in the TBPH group (Figure 1D,E, *p* < 0.05). The serum insulin content in the TBPL and TBPH groups significantly increased compared with that in the M group (Figure 1F, *p* < 0.05). Compared to the M group, the TBPH group showed significantly decreased glucagon levels, while the TBPL group presented no significant effect (Figure 1G, *p* < 0.05). The insulin resistance index in the M group was significantly higher than that in the NC group (*p* < 0.05). However, no significant difference in HOMA-IR was observed between the TBP groups and M group (Figure 1H). Additionally, the liver glycogen content in the positive control sitagliptin (SP), TBPL, and TBPH groups was significantly higher than that in the M group, while that in the SP and TBPH groups was closer to the normal level (Figure 1I, *p* < 0.05). Overall, these results indicated that TBP might be a beneficial adjuvant treatment for diabetes, and the effect of a high dose was evident.

### 2.3. Effect of TBP on Blood Glucose-Related Gut Hormones and Blood Lipid Indexes in Rats with T2DM

The secretion of cholecystokinin (CCK), peptide tyrosine tyrosine (PYY), and glucagon-like peptide-1 (GLP-1) in gastrointestinal physiology may mediate insulin secretion and blood glucose regulation [18]. The concentration levels within the NC and TBPH groups were notably elevated compared to those in the M group (Figure 2A–C, both *p* < 0.05). TBPH could stimulate insulin secretion and inhibit glucagon secretion in T2DM rats by increasing CCK, PYY, and GLP-1 to inhibit appetite, enhance satiety, prolong gastric emptying, and indirectly achieve hypoglycemic effects.

Lipid metabolism disorders are common in patients with T2DM. Figure 2D–G illustrate the lipid metabolism parameters used to evaluate the effect of TBP on dyslipidemia in T2DM rats. In the M group, a significant increase was observed in the serum total cholesterol (TC), triglyceride (TG), and low-density lipoprotein–cholesterol (LDL-C) levels, whereas a notable decrease was observed in the high-density lipoprotein–cholesterol (HDL-C) levels (*p* < 0.05), indicating a serious abnormality in blood lipid metabolism in T2DM rats. Compared to the M group, a significant decrease was observed in the TC, TG, and LDL-C levels in the TBPH group (*p* < 0.05), while the HDL-C levels saw a notable increase (*p* < 0.05). These results indicate that TBPH could regulate abnormal blood lipids in T2DM rats, among which the effect of TBPH is the most evident in reducing blood lipid levels.

### 2.4. Effect of TBP on the Viscera of T2DM Rats

Rats in the M group showed evident necrosis in the pancreatic islets, along with the infiltration of inflammatory cells compared with the NC group, according to hematoxylin and eosin (H&E) staining (Figure 3). Furthermore, the liver tissue demonstrated slight abnormalities, including increased lipid droplets, inflammatory cell infiltration, disrupted cell arrangement, mild hepatic sinusoidal dilation, and red blood cells in the hepatic sinusoids. Following supplementation with SP and TBP, the pancreatic structure showed marginal abnormalities with a tendency toward normalization. The quantity of liver lipid droplets decreased, although some residual intercellular space persisted, indicating a moderately aberrant structure with a slight enhancement from the M group. Figure 3(Af,Bf) showed that both the SP and TBPH groups had histopathological scores significantly lower than that of the M group (*p* < 0.05). Meanwhile, the histopathological score of the TBPL group decreased, but it was not significantly different from that of the M group. Overall, these results indicate that TBPH exerts a positive influence on mitigating histopathological damage in the pancreas and liver.

### 2.5. Effect of High-Dose TBP on the Composition of Gut Microbiota

Based on the physiological indexes of T2DM rats, the hypoglycemic effect of TBPH was better than that of TBPL. Subsequently, TBPH was utilized to investigate the correlation between the hypoglycemic effect of TBP on T2DM rats and the alterations in intestinal microorganisms by initially analyzing the α-diversity of the intestinal flora.

The Chao1 data showed no significant variations in the intestinal microflora diversity among the groups (Figure 4A). The species diversity was significantly increased in the TBPH group compared to that in the M group. The β-diversity was evaluated using weighted UniFrac-based principal coordinate analysis (PCoA) (Figure 4B), and the results showed a significant separation of the confidence ellipses between the M and NC groups when abundance was considered. Simultaneously, the TBPH group tended to be close to the NC group with an intersection. Thus, the intervention with TBPH significantly changed the gut microbiota structure of T2DM rats.

At the phylum level, Firmicutes and Bacteroidetes were the most abundant (Figure 5A). Linear discriminant analysis effect size (LefSe) analysis also revealed that the two phyla are potential biomarkers (Figure 5C). In the TBPH group, Firmicutes was significantly decreased (*p* < 0.05), while Bacteroidetes was significantly increased (*p* < 0.05), compared with the M group (Figure 5E–F).

Based on the relative abundance, the top five genera were *Lactobacillus*, *Prevotella*, *Bifidobacterium*, *Ruminococcus*, and *Allobaculum* (Figure 5B). The increasing trend in *Lactobacillus* and the decreasing trend in *Prevotella* in the M group could be significantly adjusted in the TBPH group. Additionally, the abundance of beneficial bacteria such as *Ruminococcus* increased to varying levels, which may be attributed to the beneficial effects of TBPH on intestinal health (Figure 5G–K, *p* < 0.05). A heatmap of the genus level revealed that *Bacteroides* and *Sutterella* also reversed the abundance of the M group to close the NC group (Figure 5D). *Bacteroides* and *Sutterella* were decreased in the M group, with their abundance significantly adjusted in the TBPH group (Figure 5L,M). These results suggest that the composition of the gut microbiota was disordered in the M group and that TBPH mitigated these intestinal dysfunctions.

### 2.6. Effect of TBPH on Untargeted Metabolome of Gut Samples

Ultrahigh-performance liquid chromatography–quadrupole time-of-flight/mass spectrometry (UPLC-QTF/MS) metabolic group analysis was used to detect non-targeted metabolites in the feces. The orthogonal partial least squares discriminant analysis (OPLS-DA) model was used to identify the potential biomarkers in the TBPH group. The scatter plot (Figure 6A,B) generated using the OPLS-DA model significantly differed among the NC, M, and TBPH groups, suggesting that TBPH significantly altered the metabolites in the intestinal tract of rats with T2DM. The volcano map visualization of differential metabolites showed that 274 metabolites were upregulated (including lupeol and ergosta-5,7,22,24(28)-tetraen-3beta-ol), while 107 were downregulated (including aflatoxin B1) in the TBPH group, compared to the M group (Figure 6C).

The Kyoto Encyclopedia of Genes and Genomes (KEGG) pathway enrichment analysis of differential metabolites identified arachidonic acid and serotonergic synapses as enhanced metabolic pathways in the TBPH group (Figure 6D).

### 2.7. Correlation Analysis among the Gut Microbiota, Metabolites, and Physiological Indexes

*Prevotella*, *Allobaculum*, and *Sutterella* positively correlated with insulin, gut hormones (GLP-1, CCK, and PYY), and hippuric acid (Figure 7). Ergosta-5,7,22,24(28)-tetraen-3beta-ol positively correlated with insulin. Regarding lipid metabolism, *Prevotella*, *Allobaculum*, *Sutterella*, and *Bacteroides* positively correlated with HDL-C and hippuric acid, and negatively correlated with TG, LDL-C, hippuric acid, and ergosta-5,7,22,24(28)-tetraen-3beta-ol.

Overall, those that had the highest correlation with physiological indexes were *Prevotella*, *Allobaculum*, and *Sutterella.* There were positive correlations among *Prevotella*, *Allobaculum*, and *Sutterella;* therefore, they may have similar functions in the gut. These results suggest that TBPH improved T2DM and may help regulate these gut microbiota and metabolism.

## 3. Discussion

Clinical and animal studies demonstrate the advantageous effects of food-derived peptides on blood glucose, fat metabolism, and intestinal function [19]. In the present study, TBP exhibited DPP-IV inhibitory effects, which might contribute to its hypoglycemic efficacy. Thus, we explored whether TBP could improve glycemic homeostasis in T2DM rats via the gut microbiota and metabolites.

The common symptoms of T2DM are hyperglycemia and impaired glucose tolerance, leading to insulin resistance [20]. In this study, sitagliptin, a DPP IV inhibitor approved by the US Food and Drug Administration in 2006 and that can prolong GLP-1 biological activities to stimulate glucose-dependent insulin secretions, was used as a positive group. TBP intervention effectively alleviated hyperglycemia, insulin resistance, and lipid metabolism disorders in T2DM rats. A previous study showed the marked effect of peptides on the release of gut hormones (CCK, PPY, and GLP-1), which prolonged the feeling of satiety, slowed gastric emptying, and promoted the secretion of insulin [21,22]. TBP prolonged the action of GLP-1 and inhibited the degradation of PYY by inhibiting DPP-IV, which plays a role in the regulation of acute energy intake, leading an improved glucose homeostasis and an increased postprandial satiety effect [23]. CCK secretion was induced by TBP, a key regulator of gastric emptying that suppresses appetite [24]. Our results are congruent with those of Caron et al. [25]. Thus, we hypothesized that TBP can stimulate CCK by activating the calcium-sensing receptor, which may also be mediated through lysophosphatidic acid receptor 5 [26,27].

TBP interventions can promote the synthesis of liver glycogen, increase the storage of glucose, and improve the disorder of glucose metabolism in T2DM. Meanwhile, lipid metabolism disorders promote fat deposition in the liver, islets, and other related complications as common metabolic disorders of T2DM [28]. In this study, the lipid metabolic parameters were determined, and the TG, TC, and LDL-C contents in T2DM rats significantly increased, while the LDL-C content decreased. Meanwhile, fat vacuoles can be seen in the liver and islet tissue, with the swelling and hypertrophy of the liver tissue. TBP intervention can reverse the above phenomenon, which is consistent with a previous study showing that treatment using ginseng peptides could improve lipid metabolism disorders in mice with T2DM [29].

Gut microbes play a crucial role in maintaining host health and are increasingly recognized as key markers for managing metabolic disorders [30,31]. In the present study, the relative abundance of Firmicutes decreased, alongside the increase in Bacteroidetes following TBP intervention, consistent with the findings of Shi et al. and Zhou et al. [32,33]. To the best of our knowledge, the presence of detrimental bacteria belonging to Firmicutes could instigate the production of endotoxins, incite inflammatory responses, and ultimately trigger insulin resistance [34]. The beneficial effects of Bacteroidetes in diabetic rats may be attributed to positive correlations with decreased host cholesterol levels [35]. This was consistent with our findings, wherein TBP increased the insulin content and improved lipid metabolism.

Additionally, significant alterations were observed at the genus level: *Prevotella*, *Ruminococcus*, *Allobaculum*, *Sutterella*, and *Bacteroides* increased, whereas *Lactobacillus*, *Coprococcus*, and *Akkermansia* decreased following TBP treatment. *Prevotella*, *Allobaculum*, *Sutterella*, and *Bacteroides* positively correlated with the insulin, gut hormone, and HDL-C levels. Increasing evidence indicates that other DPP-4 inhibitors potentially elevate the abundance of *Prevotella* [36,37]. Meanwhile, the effects of these bacteria may contribute to the gut–brain axis in mediating satiety hormones (CCK, PYY, and GLP-1), affecting feeding behavior [38] and regulating blood lipids [39]. Notably, the regulation of the intestinal flora may be the result of the joint action of multiple floras. For example, *Ruminococcus* has no significant correlation with physiological indexes, although it may affect the abundance of other flora as a dominant bacterium, thereby improving T2DM.

Several metabolites were identified after TBP intervention, among which hippuric acid and ergosta-5,7,22,24(28)-tetraen-3beta-ol were found to have important roles. Hippuric acid is positively associated with insulin secretion, gastrointestinal hormones, and HDL-C, consistent with our observation that long-term hippuric acid injections are positively associated with metabolic health, which improves glucose tolerance and increases insulin secretion [40]. In addition, due to the metabolic disorders of the intestinal flora in patients with T2DM, the level of hippuric acid negatively correlated with hyperglycemic symptoms [41]. Furthermore, ergosta-5,7,22,24(28)-tetraen-3beta-ol is positively correlated with insulin, and negatively correlated with TC, TG, and LDL-C. Ergosta-5,7,22,24(28)-tetraen-3beta-ol is a sterol intermediate involved in the ergosterol biosynthesis pathway. Its cholesterol-reducing effect may be modulated via the inhibition of cholesterol absorption and the promotion of cholesterol excretion [42]. It also contributed to the increase in vitamin D concentrations. Vitamin D plays an important role in maintaining the normal secretion of insulin [43,44].

By correlating differential metabolomics with gut microbial modulation, hippuric acid was discovered to be negatively correlated with *Lactobacillus* and positively correlated with *Sutterella*. A previous study showed that hippuric acid was metabolized by *Lactobacillus* [45]. Zhang et al. showed that the GLP-1 agonist led to a significant increase in the relative abundance of *Sutterella* against T2DM [46]. It could be inferred that TBP as a DPP-IV inhibitor prolongs the action of GLP-1, increases the abundance of *Sutterella*, and promotes the secretion of hippuric acid. Furthermore, ergosta-5,7,22,24(28)-tetraen-3beta-ol was negatively correlated with *Lactobacillus*, similar to the findings of a previous study [47]. Overall, these findings imply that TBP has a potential protective role in T2DM through ameliorating intestinal microbial imbalances and influencing metabolites.

## 4. Materials and Methods

### 4.1. Materials

Two-year-old cultured *T. bimaculatus* were purchased from Zhangzhou Senhai Food Co., Ltd. (Zhangzhou, China). The fish skin was ground using a meat grinder.

Alkaline protease (EC3.4.21.62) was purchased from SolarBio (Beijing, China). The plasma glucagon, insulin, TG, TC, LDL-C, HDL-C, GLP-1, CCK, and PYY levels were obtained from the Jiancheng Bioengineering Institute (Nanjing, China). Unless otherwise stated, all the other chemicals were of analytical grade.

### 4.2. Preparing TBSH

The hydrolysate of TBSH was prepared as previously described [21]. Briefly, TBSH underwent treatment with distilled water (10% *w*/*v*) at 50 °C and a pH of 9. Alkaline protease (8000 U/g) was used for enzymatic hydrolysis over a duration of 4 h, followed by 20 mins of boiling to deactivate the enzyme. Subsequently, the solution was cooled to 25 °C and centrifuged at 10,000× *g* for 15 min at 4 °C. The resulting product was subsequently freeze-dried and preserved at −20 °C for future analysis.

### 4.3. Distribution of TBSH Molecular Weight

The molecular weight distribution of TBSH underwent high-performance gel-filtration chromatography using a Toyo Soda TSKgel G2000 SWXL column. A 10 μL of sample was injected using the mobile phase buffer acetonitrile/water/trifluoroacetic acid (45:55:0.1%, *v*/*v*) at a flow rate of 0.5 mL/min; the column temperature was 30 °C. The wavelength range used for the detection was 214 nm. Gly-Gly-Gly, Gly-Gly-Tyr-Arg, bacitracin, aprotinin, and cytochrome C were used as standards. A standard curve was established according to GBT 22729-2008 [48].

### 4.4. DPP-IV Inhibitory Activity In Vitro

TBSH was graded using an ultrafiltration membrane, and the obtained components were marked as TBP (Mw < 1 kDa) or TBP1 (Mw ≥ 1 kDa). The two components and TBSH were analyzed for DPP-IV inhibitory activity in vitro using a previously described method, with a modification [49]. DPP-IV (40 μL) was mixed with 10 μL of TBSH samples in a 96-well plate and incubated at 37 °C for 10 min. Subsequently, 50 μL of DPP-IV substrate (G-P-AMC) was added and incubated for an additional 30 min. The absorbance of each well was measured at the excitation and emission wavelengths of 360 and 460 nm, respectively. DPP-IV inhibition was calculated using Equation (1):(1)$$Inhibition\left(\%\right)=\frac{\left(A_{0}-A_{i}\right)}{A_{0}}\times 100$$
where *A*_0_ is the absorbance of the control and *A_i_* is the absorbance of the sample.

The IC_50_ value was defined as the peptide concentration of a sample that inhibited 50% of the DPP-IV activity under the assay conditions.

### 4.5. Animal Experiments

Fifty male SD rats (6 weeks old) were purchased from Shanghai Meixuan Biotechnology Co., Ltd. (Shanghai, China) and maintained at 23–25 °C, 60% relative humidity, and under a 12 h light/dark cycle. One week following adaptation, six randomly separated rats were fed a normal, laboratory standard, pelleted diet as the normal control (NC) group. The other rats were administered 30 mg/kg streptozotocin (STZ) to establish a diabetic rat model. Seventy-two hours after withdrawal of the drug, a glucose concentration of ≥16.7 mmol/L was considered to have led to the successful establishment of a T2DM rat model. The twenty-four diabetic rats were then randomly assigned to four groups (*n* = 6): M, SP (sitagliptin 20 mg/kg), TBPL (200 mg/kg), and TBPH (400 mg/kg). Throughout the experiment, water and food were provided ad libitum, with energy ratios of 12% fat, 66% carbohydrates, and 22% protein. The body weight and fasting blood glucose levels were measured weekly. Following a four-week period, cervical dislocation was performed after an overnight fast. Blood samples were collected from the abdominal aorta, the coagulation of whole blood was performed with a clotting vessel, and serum was collected by centrifugation. Liver and pancreatic tissues were isolated, and the intestinal contents and intestines were promptly frozen at −80 °C for further analysis. All procedures involving rats were conducted in compliance with the protocol approved by the Ethics Committee of Jimei University in China (approval no. JMULAC201159), adhering to the guidelines outlined by the European Community on the care and utilization of laboratory animals (Directive 2010/63/EU).

### 4.6. Oral Glucose Tolerance Test (OGTT)

The rats were subjected to an overnight fasting period and subsequently administered a 25% glucose solution at a dosage of 1.5 g/kg. The blood glucose levels were monitored at 0, 0.5, 1, and 2 h post-glucose administration. A blood glucose response curve was constructed for each group based on the recorded glucose values, and the AUC was determined within 2 h [50].

### 4.7. Blood Glucose-Related Gut Hormones and Blood Lipids

The levels of insulin, glucagon, hepatic glycogen, CCK, PYY, and GLP-1 were measured using enzyme linked immunosorbent assay (ELISA) kits [51]. The serum TC, TG, HDL-C, and LDL-C levels were determined using an automatic biochemical analyzer [52].

### 4.8. Liver and Pancreas Histological Analysis

Tissues were fixed with 4% paraformaldehyde, dehydrated, embedded in paraffin, and sliced. Histological changes were observed using H&E staining and light microscopy (Shanghai Meixuan Biotechnology Co., Ltd., Shanghai, China). The severity of liver injury was assessed by Suzuki’s criteria [53]. Histopathological scores were given as follows: 1 = individual cellular edema, vacuolization, or single cellular necrosis; 2 = little cellular edema, vacuolization, or necrosis ≤ 30%; 3 = moderate cellular edema, vacuolization, or necrosis ≤ 60%; and 4 = severe edema, vacuolization, and necrosis >60%.

### 4.9. Gut Microbiota Analysis

The intestinal contents of the rats were submitted to Panomix Biomedical Technology Co., Ltd. (Suzhou, China) for 16S rRNA gene sequencing. The analysis included genome extraction, PCR amplification and DNA purification, and the data analysis included species annotation, α-diversity analysis, β-diversity analysis, and species difference analysis [54].

The V4 region of the bacteria 16S rRNA gene was amplified by primers 341F ATGCGTAGCCGACCTGAGA and 805R CGTCAGACTTTCGTCCATTGC. The products were quantified by 2% agarose gel electrophoresis and fluorescence electrophoresis. The sample barcode label sequence was identified and segmented to obtain analytical data. After less than 20 base pairs were removed from the tail of reads, data filtering quality control was performed to obtain valid data. Quantitative Insights into Microbial Ecology (QIIME2) software (2019.4, The Broad Institute of MIT and Harvard, Cambridge, MA, USA) was utilized for quality control splicing, filtering, and other preprocessing of the raw data, and Usearch software (version 11, ROTA4 Corporation, Omaha, NE, USA) was used to conduct a cluster analysis of the Operational Taxonomic Units (OTUs) and a classification of the community species, and annotate the OTU species [55]. The α-diversity was evaluated using the Chao index and Shannon index [56]. Partial least square discriminant analysis was employed to illustrate β-diversity. The taxonomic composition of the community at the phylum level is visualized with a bar plot. To identify the dimensional gut bacteria and characterize the microbial differences between different groups, LefSe analysis was performed [57]. Differences were considered statistically significant if *p* < 0.05 and the linear discriminant analysis (LDA) score was ≥ 4.

### 4.10. Fecal Metabolome Analysis

The intestinal contents of the rat feces were submitted to Panomix Biomedical Tech Co., Ltd. (Suzhou, China) for non-targeted metabolome group analysis. The ultra-high-performance liquid phase system (Waters, Milford, MA, USA) uses an ACQUITY UPLC HSS T3 column for detection, as previously described [58]. The column was maintained at 40 °C. The flow rate and injection volume were set at 0.3 mL/min and 2 μL, respectively. In positive ion mode, the mobile phases consisted of (A) 0.1% formic acid in water (*v*/*v*) and (B) 0.1% formic acid in acetonitrile (*v*/*v*). In negative ion mode, the mobile phases consisted of (A) ammonium formate (5 mM) and (B) acetonitrile. The chromatographic gradient elution procedure was as follows: 0–1 min, 8% B; 1–8 min, 8–98% B, 8–10 min, 98% B; 10–10.1 min, 98–8% B; 10.1–12 min, 8% B.

The LC-MS/MS data were converted to mzXML file format by the Proteowizard software (v3.0.8789, Julich Phenomenal in collaboration with VU University Amsterdam, Amsterdam, The Netherlands) [59]. The RXCMS software (v1.2.3, Rackware, Guildford, UK) package was used for peak detection, peak filtering and peak alignment [60]. Support vector regression correction based on QC samples was used to eliminate systematic errors. Substances with Coefficient of Variance (CV) less than 30% in the QC samples were then retained for subsequent analysis [61]. The sample data were subjected to orthogonal partial squares discriminant analysis (OPLS-DA), and volcano plots were generated to demonstrate the differences in the metabolite composition among individual samples. The *p*-value was calculated according to the statistical test, and the variable importance in the projection (VIP) was calculated using the OPLS-DA. Metabolites were considered to be significant when the *p* < 0.05 and the VIP > 1.

### 4.11. Statistical Analysis

IBM SPSS Statistics for Windows(v19, IBM, Armonk, NY, USA), was used to analyze the experimental data. Data are expressed as mean ± standard error of the mean (SEM), *n* = 6. Multiple comparisons were made using one-way analysis of variance (ANOVA), and differences among groups were compared using the Duncan Multiple Range Test. A *p*-value of <0.05 indicated significant differences.

## 5. Conclusions

TBP exhibited DPP-IV inhibitory activity that had a significant hypoglycemic effect and alleviated lipid metabolism disorders in T2DM rats. Furthermore, it positively impacted the composition of the intestinal microbiota by elevating *Prevotella*, *Allobaculum*, and *Sutterella* while diminishing *Lactobacillus*. Significantly, metabolites such as hippuric acid and ergosta-5,7,22,24(28)-tetraen-3beta-ol are related to gut microbiota and diabetes-related indexes. TBP may be an effective and nutritional supplement in T2DM rats by modulating the gut microbiota and their metabolites. TBP is a promising intervention for diabetes prevention or treatment. Nevertheless, it also suffers from insufficient in vivo data including the absence of dosage, pharmacokinetic data and variabilities in intake; in addition, dietary peptide–drug interactions are issues because these can lead to variable biological effects that need to be further elucidated in vivo. Furthermore, the mechanism of TBP should be examined in vitro and in vivo using metagenomics and targeted metabolomic approaches.

## Figures and Tables

**Figure 1 marinedrugs-22-00377-f001:**
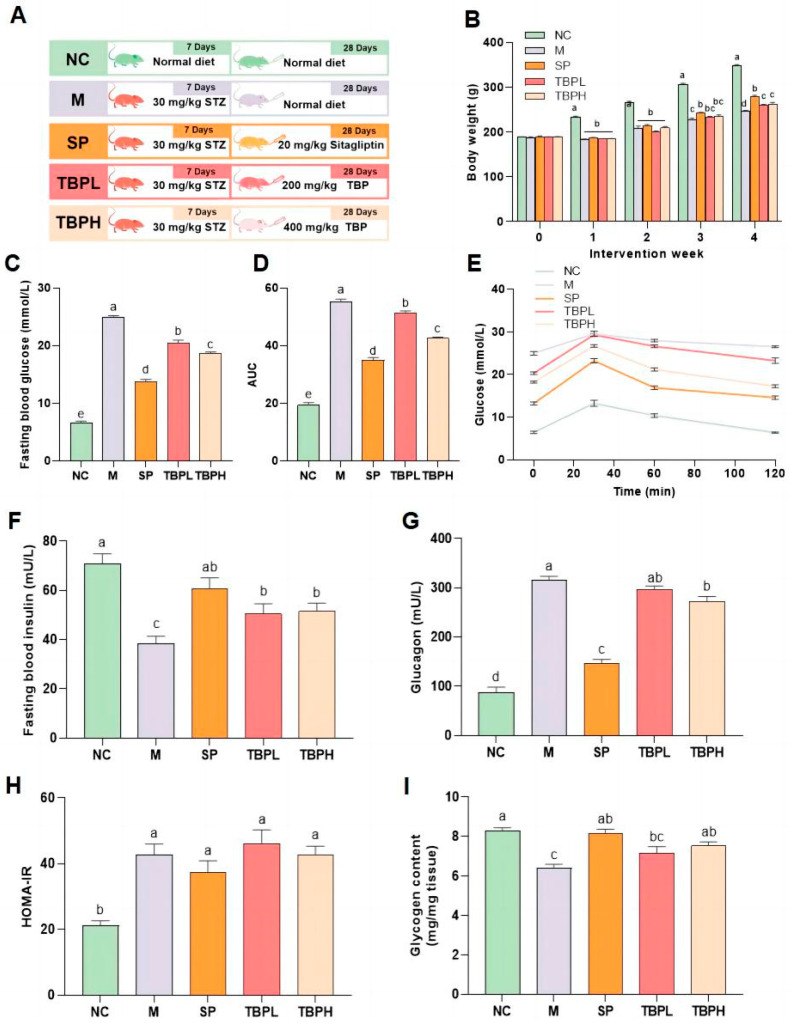
Effects of <1 kDa peptide fraction from Takifugu bimaculatus fish skin hydrolysate (TBP) on body weight and basic indices in rats with T2DM. (**A**) Experimental protocol; (**B**) body weight; (**C**) fasting blood glucose, (**D**) oral glucose tolerance test, and (**E**) area under curve, AUC; (**F**) Fasting blood insulin; (**G**) glucagon; (**H**) homeostasis model assessment of insulin resistance, HOMA-IR; (**I**) glycogen. Results are expressed as the mean ± standard error of the mean (SEM) (*n* = 6). Different letters represent significant differences (*p* < 0.05) between groups using one-way analysis of variance (ANOVA) and Duncan’s multiple-range test. The same letters indicate *p* > 0.05.

**Figure 2 marinedrugs-22-00377-f002:**
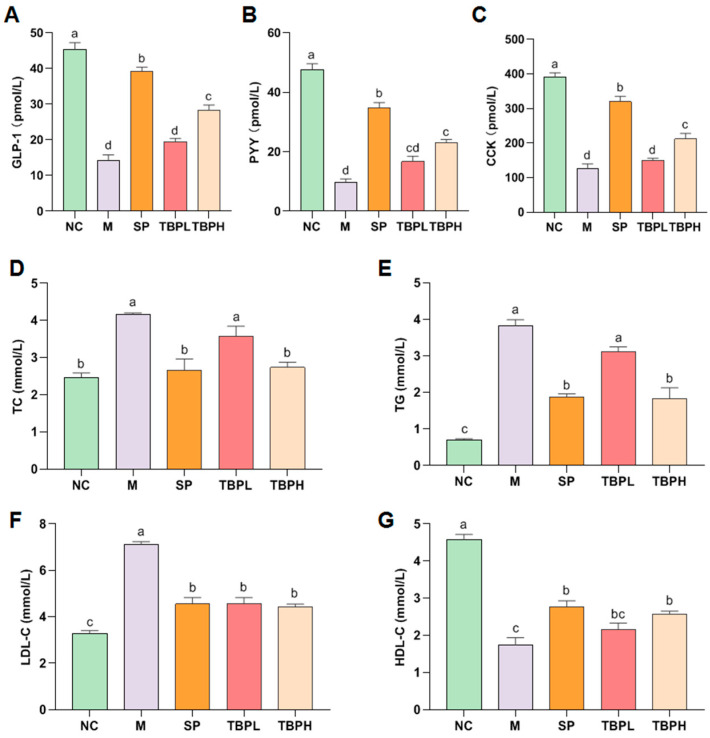
Effect of TBP on blood-glucose-related hormones and blood lipid metabolism in rats with T2DM. (**A**) glucagon-like peptide-1, GLP-1; (**B**) peptide tyrosine tyrosine, PYY; (**C**) cholecystokinin, CCK; (**D**) total cholesterol, TC; (**E**) triglyceride, TG; (**F**) low-density lipoprotein cholesterol, LDL-C; (**G**) high-density lipoprotein cholesterol, HDL-C. Results are expressed as the mean ± standard error of the mean (SEM) (*n* = 6). Different letters represent significant differences (*p* < 0.05) between groups using one-way analysis of variance (ANOVA) and Duncan’s multiple-range test. The same letters indicate *p* > 0.05.

**Figure 3 marinedrugs-22-00377-f003:**
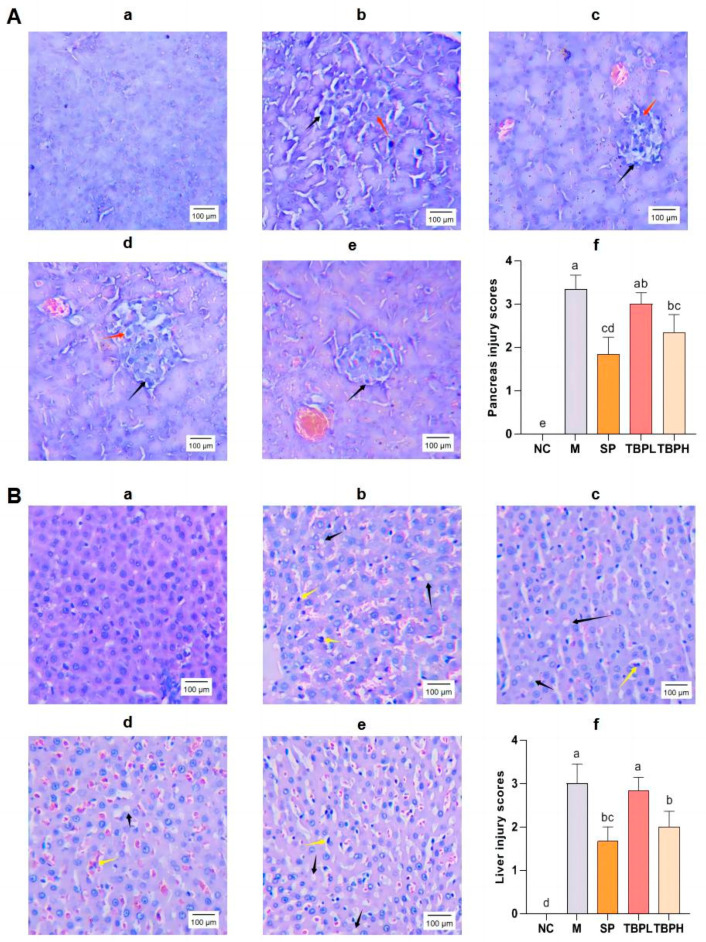
TBP improves the histopathological characteristics of rats with T2DM. (**A**) Hematoxylin and eosin (H&E) stained images of the pancreas taken with a Leica microscope (200×, scale bar, 100 μm); the black arrows indicate inflammatory cells and red arrows show necrotic islet β cells. (**a**) NC group, (**b**) M group, (**c**) SP group, (**d**) TBPL group, (**e**) TBPH group, and (**f**) the injury scores for the pancreas. (**B**) Hematoxylin and eosin (H&E) stained images of the liver taken with a Leica microscope (200×, scale bar, 100 μm); the black arrows indicate lipid droplets and yellow arrows show inflammatory liver cells. (**a**) NC group, (**b**) M group, (**c**) SP group, (**d**) TBPL group, (**e**) TBPH group, and (**f**) the injury scores of liver. Results are expressed as the mean ± standard error of the mean (SEM) (*n* = 6). Different letters represent significant differences (*p* < 0.05) between groups using one-way analysis of variance (ANOVA) and Duncan’s multiple-range test. The same letters indicate *p* > 0.05.

**Figure 4 marinedrugs-22-00377-f004:**
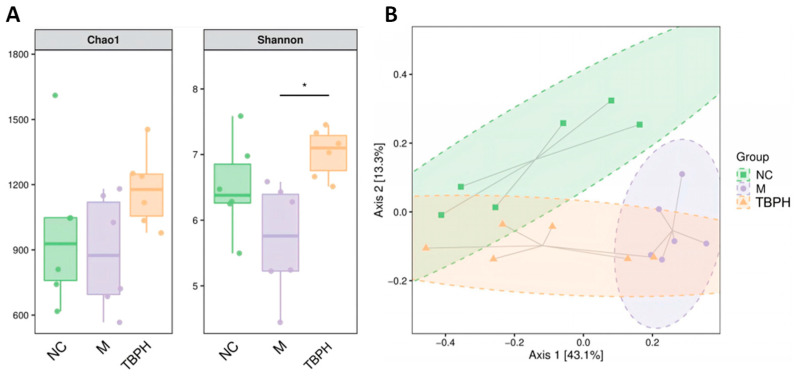
Effect of TBPH on the composition of the gut microbiota. (**A**) Chao1 and Shannon indices and (**B**) principal coordinate analysis (PCoA). The statistically significant correlation between two variables was tested using the Kruskal-Wallis one-way ANOVA test, * *p* < 0.05.

**Figure 5 marinedrugs-22-00377-f005:**
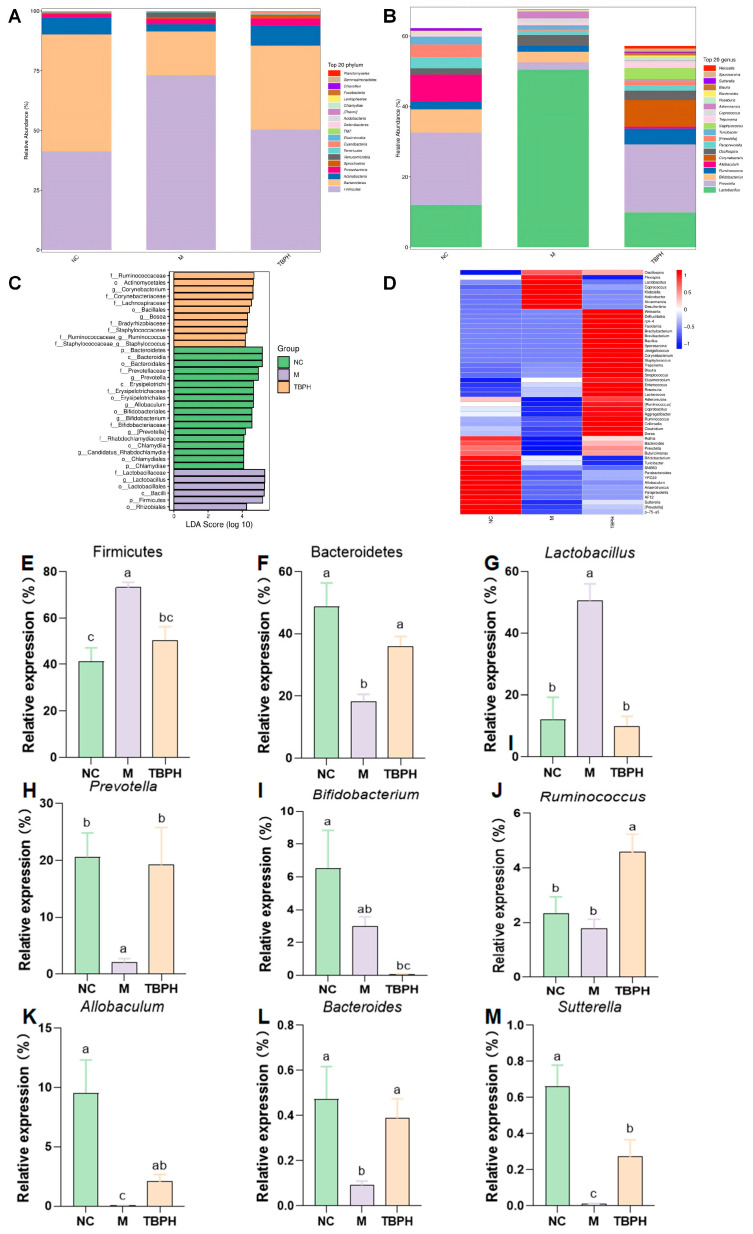
Effect of TBH on the gut sample composition. (**A**) Relative abundance at the phylum level and (**B**) genus level; (**C**) LDA, linear discriminant analysis, LDA score > 4; (**D**) Heatmap of bacterial distribution at the genus level; (**E**–**M**) the relative abundance of key bacteria at the phylum and genus level. Results are expressed as the mean ± standard error of the mean (SEM) (*n* = 6). Different letters represent significant differences (*p* < 0.05) between groups using one-way analysis of variance (ANOVA) and Duncan’s multiple-range test. The same letters indicate *p* > 0.05.

**Figure 6 marinedrugs-22-00377-f006:**
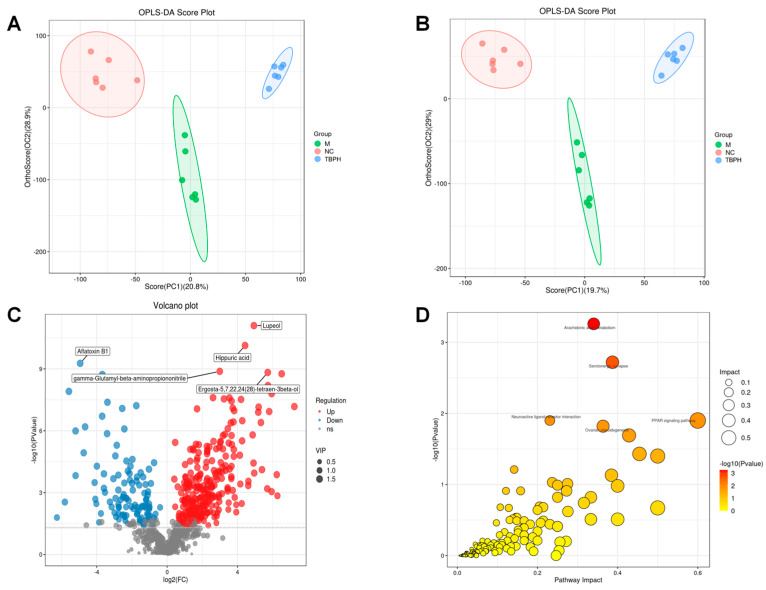
Effect of TBH on untargeted metabolomics in gut samples. (**A**) Score plots of OPLS-DA in the positive modes and (**B**) negative modes; (**C**) volcano plot; (**D**) metabolic pathway enrichment analysis.

**Figure 7 marinedrugs-22-00377-f007:**
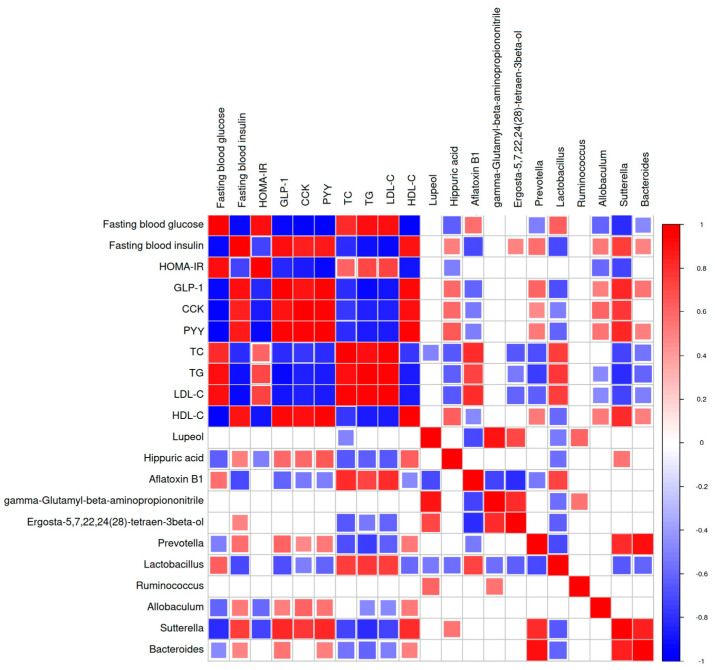
Spearman’s correlation heatmap between the gut microbiota, metabolites, and the biochemical parameters. The statistically significant correlation between two variables was tested using the Pearson correlation coefficient, *p* < 0.05.

## Data Availability

The original contributions presented in the study are included in the article/Appendix A, further inquiries can be directed to the corresponding authors.

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
