# Peer review of "Mechanism of *Takifugu bimaculatus* Skin Peptides in Alleviating Hyperglycemia in Rats with Type 2 Diabetic Mellitus Based on Microbiome and Metabolome Analyses"

_marinedrugs, 2024, doi:10.3390/md22080377_

Round 1

Reviewer 1 Report

Comments and Suggestions for Authors

This study investigated the potential antidiabetic effects of the hydrolysate on T. bimaculatus skin. The hypoglycemic mechanism of this substance was confirmed based on the intestinal - flora metabolism. TBP may have potential as a supplement for the treatment of type 2 diabetes. However, I would like to bring the author's attention to certain aspects that require further improvement in order to enhance clarity for readers.

1. In the results section 2.2, it is recommended to consolidate each paragraph and include a concluding sentence at the end of each paragraph to enhance clarity for readers.

2. In the results section 2.2, the statement "Conversely, the TBH group showed a downward trend; however, no significant difference was observed compared with the M group" is contradictory to the data provided, as no downward trend was observed.

3. In the discussion section, the statement "Type 2 diabetes mellitus manifests through hyperglycemia and insulin resistance, which are reversed by treatment with TBP. It is usually accompanied by abnormal hormone levels" is unclear in its meaning.

4. The authors should explain why this study used sitagliptin as positive agents.

5. The discussion section merely summarizes the results and lacks a more in-depth analysis. The filtered path analysis could be appropriately supplemented.

6. Other researchers should replicate the experimental methods described in the paper. Additionally, the authors should provide a more detailed description of the metabolic analysis process in the experimental method section.

Comments on the Quality of English Language

Overall, the grammar and fluency of the article are satisfactory; however, certain transitions appear to be abrupt. Therefore, it is recommended to thoroughly review and refine the entire text.

Reviewer 2 Report

Comments and Suggestions for Authors

In this manuscript, the authors have studied the effect of Takifugu bimaculatus skin peptides against hyperglycemia. The mechanism was studied based on microbiome and metabolomic Analysis. Some issues should be clarified here:

(1) In line 26, the keywords “non-targeted metabolic” should be changed to “non-targeted metabolome.

(2) In line 52, the authors mentioned that “Collagen hydrolysates are rich in proline and alanine, which are potential DPP-IV inhibitors”, please add the reference. Please determine the amino acid composition of TBP and clarify the relationship between its amino acid composition and DPP-IV inhibition activity. Please summarize the characteristics of DPP-IV inhibition peptides.

(3) In line 60, camel milk was illustrated, while it is not closely related with peptide.

(4) In line 84, the groups of TBL and TBH need to be named as TBPL and TBPH.

(5) In line 104 Figure 1A, the units should be changed as mg/kg. Besides, in Figure 1D, the results of AUC need to be presented as big picture similar to Figure 1C.

(6) In line 198, arrows should be added to the specific histopathological characteristics changes. Besides, the HE staining should be scored. In addition, the fat area should be stained with oil-red O and quantified.

(7) How can the authors believe that their T2DM rats model is successfully established? Please presented the standards of T2DM here.

(8) Why the authors only studied the microbiome and metabolome of NC, T2DM and TBH groups? Please showed the significant changed bacteria at phylum and genus level with histograms.

(9) In the discussion part, the relationship between microbiome results and metabolome results was not clarified clearly. Please discuss this part in depth.

(10) In line 401, why the authors only determined 6 rats in each group?

Comments on the Quality of English Language

Minor editing of English language required.

Reviewer 3 Report

Comments and Suggestions for Authors

See the attached review report.

Reviewer 4 Report

Comments and Suggestions for Authors

This study on Takifugu bimaculatus skin peptides (TBP) as a potential dietary supplement for type 2 diabetes management is comprehensive and innovative. However, several areas require clarification and improvement:

a) Methodology:

  • Justify the choice of Takifugu bimaculatus as the peptide source. (Lines 29-35)
  • Explain the rationale behind the doses (200 mg/kg and 400 mg/kg) used in the animal study. (Lines 83-85)
  • Provide details on the selection of alkaline protease for enzymatic hydrolysis. (Lines 324-328)
  • In the DPP-IV inhibitory activity assay, include a dose-response curve rather than testing a single concentration. (Lines 339-349)

b) Results interpretation:

  • Address the fluctuations in fasting blood glucose levels in the TBH group (Figure 1C). (Lines 87-94)
  • Provide quantitative analysis for the histological results (Figure 4), possibly using a scoring system. (Lines 144-154)
  • Elaborate on the statistical significance of the gut microbiota changes, including correction for multiple comparisons. (Lines 279-288)

c) Discussion:

  • Compare TBP's effects on liver glycogen content and histological improvements to other known antidiabetic interventions. (Lines 111-117 and 144-154)
  • Explain the potential mechanisms behind the increase in gut hormones (CCK, PYY, GLP-1) after TBP treatment. (Lines 258-262)
  • Discuss how the observed changes in gut microbiota composition might contribute to TBP's hypoglycemic effects. (Lines 279-288)
  • Elaborate on the potential mechanisms by which identified metabolites (hippuric acid and ergosta-5,7,22,24(28)-tetraen-3beta-ol) might influence glucose metabolism and insulin secretion. (Lines 302-311)

d) Future directions:

  • Address potential side effects and safety concerns of TBP as a nutritional supplement. (Lines 45-51)
  • Discuss plans for human studies and how findings will be applied to human subjects. (Consider adding this to the conclusion section)
  • Outline future research to study the molecular mechanisms of action in more detail. (Consider adding this to the conclusion section)

Comments on the Quality of English Language

The overall quality of English in the manuscript is good. However, there are some minor grammatical errors and awkward phrasings that should be addressed to improve clarity and readability. A thorough proofreading by a native English speaker is recommended.

Round 2

Reviewer 4 Report

Comments and Suggestions for Authors

Your dedication to addressing the previous concerns and suggestions is evident, and the changes you have made have substantially improved the manuscript. The enhanced clarity, coherence, and scientific rigor are particularly noteworthy. Your careful integration of the recommended modifications has significantly elevated the quality and impact of the work. At this stage, I have no further substantive comments, as the manuscript now meets the high standards required for publication. I fully support its acceptance.